# Revealing the Development Patterns of the Mandibular Glands of *Apis mellifera carnica* Based on Transcriptomics and Morphology

**DOI:** 10.3390/insects15030176

**Published:** 2024-03-06

**Authors:** Chunlei Pan, Yi Zhang, Chunguang Liu, Zhihao Zhang, Liang Tao, Kang Wang, Zheguang Lin, Ting Ji, Fuchao Gao

**Affiliations:** 1Mudanjiang Branch of Heilongjiang Academy of Agricultural Sciences, Mudanjiang 157043, China; 101pcl@163.com (C.P.); mdjnkybgs@126.com (C.L.); talent-zzh@163.com (Z.Z.); taoliangnky@163.com (L.T.); 2College of Animal Science and Technology, Yangzhou University, Yangzhou 225009, China; zy162207123@163.com (Y.Z.); kwang@yzu.edu.cn (K.W.); z.lin@yzu.edu.cn (Z.L.)

**Keywords:** *Apis mellifera carnica*, mandibular gland, royal jelly, single-male insemination, transcriptome

## Abstract

**Simple Summary:**

The mandibular gland is one of the main glands used by worker bees to synthesize and secrete royal jelly. To investigate the development patterns of the mandibular gland and the molecular regulatory mechanisms involved in royal jelly secretion, we dissected the mandibular glands of *Apis mellifera carnica* worker bees of different ages (3, 6, 9, 12, and 16 d) and performed morphological and transcriptomic analyses. Microscopy revealed that the mandibular glands likely completed development in the early stages. There were no significant differences in the structural morphology or organelles involved in the secretion of royal jelly at different ages. Although the condition of the mandibular glands from 3 to 16 d could not be classified based on morphological changes, transcriptomic data showed that the process of royal jelly secretion by the mandibular gland (3–16 d) could be divided into three stages with different expression patterns: 3–6 d, 9–12 d, and 16 d. These results elucidate the genetic basis of the involvement of the mandibular gland in the process of royal jelly secretion in *Apis mellifera*.

**Abstract:**

The mandibular gland in worker bees synthesizes and secretes the organic acids present in royal jelly, and its development directly affects yield and quality. Therefore, we aimed to analyze the differences in morphology and gene expression in the mandibular glands of *Apis mellifera carnica* worker bees of different ages (3, 6, 9, 12, and 16 d). We dissected their mandibular glands and performed morphological and transcriptomic analyses to investigate the development of the mandibular gland and the molecular regulatory mechanisms involved in royal jelly secretion. Microscopy revealed that mandibular gland development is likely completed in the early stages. There were no significant differences in the structural morphology or organelles involved in the secretion of royal jelly at different ages. Transcriptomics revealed a total of 1554 differentially expressed genes, which were mainly involved in fat metabolism, lipid transport, and energy metabolism. The extracellular matrix–receptor interaction pathway was significantly enriched and contributed to the royal jelly secretion process. These results elucidate the genetic basis of the role of the mandibular gland in royal jelly secretion in *A. mellifera* and provide a reference for the genetic improvement of bees with high royal jelly production in the future.

## 1. Introduction

Royal jelly is a light-yellow, milky substance secreted by multiple glands, such as the hypopharyngeal and mandibular glands, in the head of a worker bee [1]. It is mainly used to feed one- to three-day-old larvae and queen bees in colonies and plays a key role in the caste differentiation of bees [2]. Royal jelly contains various biologically active substances and is rich in proteins, sugars, and lipids and contains small quantities of free amino acids, vitamins, and minerals [3]. The active substances in royal jelly have certain antioxidant [4,5], anti-inflammatory [6,7], antibacterial [8], antiviral [9], antitumor [10,11], neuroprotective [12], and anti-aging [13] properties. Consequently, royal jelly has been widely recognized by consumers as a natural functional health food. Compared with other bee products, such as honey, royal jelly has higher economic value, and its demand is increasing each year. Over the past 40 years, scientific researchers and individual beekeepers in the Jiangsu and Zhejiang areas of China have continuously selected bees with high royal jelly production and improved royal jelly collection technology. Royal jelly production from a single colony of Western honeybees has been greatly improved; one colony of high royal jelly production bees can provide more than 10kg of royal jelly per year, which is more than 10 times that of *A. mellifera ligustica* [14].

To date, many studies have attempted to elucidate the basis of royal jelly secretion by worker bees using various approaches, including transcriptomics, genomics, proteomics, and metabolomics [14,15,16,17,18,19]. Previous studies mainly focused on the hypopharyngeal glands, which secrete the protein components of royal jelly [20]; however, few studies have been conducted on the involvement of the mandibular gland in royal jelly synthesis and secretion. Previous studies have shown that the mandibular gland mainly synthesizes and secretes fatty acids used for larval nutrition and pheromones related to bee colony alarm signals [21,22]. The fatty acids secreted by the mandibular gland include ω-hydroxy acids and their corresponding diacids [23], among which 10-hydroxy-2-decenoic acid (10-HDA) is a unique unsaturated fatty acid that accounts for approximately 1.4–1.8% of the wet weight of royal jelly. Owing to its unique stability, 10-HDA is a recognized indicator for evaluating royal jelly quality [24]. A transcriptomic-based comparative analysis of the synthesis pathways of the mandibular gland secretions of queen and worker bees found that the cytochrome P450 (CYP450) family genes in the mandibular gland are involved in fatty acid β-oxidation and ω-oxidation [25]. Proteomic analysis of the mandibular glands of *A. mellifera* has shown that specific and highly abundant proteins are mainly enriched in pathways related to transport and lipid synthesis [26]. Non-targeted lipidomic analysis of newly emerged, 6 and 15 d old worker bees revealed that lipids in the mandibular gland differed dramatically across developmental stages [27]. However, the existing research is insufficient with respect to fully elucidating the molecular mechanisms of the mandibular gland in the process of royal jelly secretion by worker bees.

In this study, we collected 750 *A. m. carnica* worker bees of five different ages, dissected their mandibular glands, and used electron microscopy combined with transcriptomics to analyze the differences in the morphology and gene expression of their mandibular glands. We aimed to reveal the changes in the mandibular glands of worker bees of different ages during royal jelly secretion. Through annotation and enrichment analysis of the differentially expressed genes (DEGs) at different ages, we identified the key signaling pathways and genes involved in the secretion of royal jelly from the mandibular gland. The results of this study enhance our understanding of royal jelly secretion and provide a reference for breeding improved *A. mellifera* specimens.

## 2. Materials and Methods

### 2.1. Sample Collection

The six single-male-inseminated queen bees (*A. m. carnica*) used in this study were provided by the National Bee Gene Bank (Jilin, China), and the six bee colonies were organized by the Experimental Bee Farm of Yangzhou University. Before the start of the experiment, the bee colonies were adjusted uniformly; sufficient sugar, water, and pollen were provided during the breeding period, and sources of rapeseed nectar were sporadically present in the area. The backs of the newly emerged worker bees were marked, and the day of emergence was recorded as 0 d. Worker bees that were 3, 6, 9, 12, and 16 d old were collected. A total of 150 bees were collected at each age, of which 30 were used for scanning electron microscopy, another 30 for transmission electron microscopy, and 90 for transcriptome sequencing. Each sample used for transcriptome studies was a pool of 30 bees. After sampling, the mandibular glands were quickly dissected under a stereomicroscope and placed in 2.5% glutaraldehyde or liquid nitrogen for subsequent use.

### 2.2. Microscopy

We utilized scanning electron microscopy and transmission electron microscopy to observe the tissue structure of the mandibular gland. The bees were placed on a 4 °C wax plate, and their mandible tissue (connected to the mandibular glands) was quickly dissected under a microscope. Other aponeurosis tissue was peeled off to obtain a complete gland. Subsequently, the glands were post-fixed in osmium tetroxide. After dehydration, drying, and conductive treatment, images were captured using a scanning electron microscope. The sample preparation and fixation processes for transmission electron microscopy were consistent with those used for the scanning electron microscopy method. However, the mandibular gland tissue had to be embedded in resin after dehydration. After staining, the sections were examined using transmission electron microscopy.

### 2.3. Library Construction and Transcriptomic Sequencing

For transcriptome sequencing, 3 μg of RNA was used per sample. Sequencing libraries were generated using the NEBNext^®^ UltraTM RNA Library Prep Kit for Illumina^®^ (NEB, Ipswich, MA, USA). The reverse transcription quantitative polymerase chain reaction (RT-qPCR) products were purified using the AMPure XP system (Beckman Coulter, Brea, CA, USA), and library quality was assessed using the Agilent 2100 Bioanalyzer system (Agilent, Santa Clara, CA, USA). The library was sequenced using an Illumina NovaSeq 6000 platform (Novogene Biotechnology Co., Ltd., Beijing, China), and 150 bp paired-end reads were generated. The quality of RNA sequences was evaluated using FastQC (https://www.bioinformatics.babraham.ac.uk/projects/fastqc/) (accessed on 11 July 2021), and sequence adapters and low-quality reads (read quality < 20) were removed. The sequenced reads were mapped to the reference genome using HISAT v2.1.0 [28]. To quantify the expression of each transcript, the alignment results were analyzed using FeatureCounts (v1.5.0-p3) software [29].

### 2.4. DEG Identification and Analysis

Following the standard consisting of a *p*-value ≤ 0.05 and |log2(Fold change)| ≥ 1, the DEGs among worker bees of different ages were screened using DESeq2 [30]. The resulting *p*-values were adjusted using the Benjamini–Hochberg method to control for the false discovery rate. Gene Ontology (GO) and Kyoto Encyclopedia of Genes and Genomes (KEGG) enrichment analyses were performed using the ‘clusterProfiler’ R package (v4.0) [31]. Results with corrected *p*-values of less than 0.05 were considered significantly enriched. Venn and expression cluster analyses were performed using the OmicShare platform (https://www.omicshare.com/) (accessed on 10 November 2023).

### 2.5. RT-qPCR Validation of DEGs

Fifteen DEGs (GB46290, GB45151, GB47301, GB51787, GB45400, GB42616, GB53753, LOC102656448, GB47694, GB47327, GB41227, GB40240, GB47303, GB42431, and GB41159) were randomly selected and subjected to RT-qPCR validation. The β-actin gene was used as an internal reference. The relative gene expression was calculated based on the 2^−ΔΔCt^ method.

## 3. Results

### 3.1. Mandibular Gland Morphology

The surface morphology of the mandibular gland was observed using scanning electron microscopy, and images were collected at different angles (Figure 1). The mandibular glands were “V”- or heart-shaped, with rough and wrinkled surfaces, and many ducts and secretory points were observed on the surface. There were almost no differences in the surface morphology of the mandibular glands at different ages, indicating that the development of the mandibular glands may be completed at an early stage.

Transmission electron microscopy (Figure 2) revealed that the morphology of the mandibular gland cells at each age was relatively regular, with clear cell membranes and no obvious damage. A large number of oval or elongated mitochondria, a small number of lysosomes, multiple intracellular transport ducts, and abundant rough endoplasmic reticula (RER) in an expanded vesicular state were observed at each age. Local shedding of ribosomes occurred on the surface of the RER. In addition, a small number of autophagolysosomes were observed at 6 and 9 d, and those at 9 d contained undigested mitochondria and RER.

### 3.2. Quality Control and RNA Sequencing Data Alignment

A total of 850,235,782 raw reads were obtained by sequencing the mandibular gland transcriptomes of worker bees of five different ages. After filtering, 823,548,318 clean reads were obtained, totaling 123.52 Gb. The Q20 and Q30 values were above 96.89% and 91.61%, respectively, indicating acceptable quality (Table 1). After comparing the sequencing data with the reference genome, the average mapping rate was 78.14%, and the reads aligned to the exon region accounted for 90.33–93.75% (Appendix A). Quality control and alignment showed that the transcriptome-sequencing data complied with the standards and could be used for subsequent analyses.

### 3.3. Differential Gene Expression

Using AmC-3 as the control group, the numbers of upregulated genes were 256 (AmC-6 vs. AmC-3), 212 (AmC-9 vs. AmC-3), 291 (AmC-12 vs. AmC-3), and 370 (AmC-16 vs. AmC-3), while those of the downregulated genes were 197, 285, 413, and 776, respectively (Figure 3). Additionally, a total of 166 shared genes (10.68%) were identified (Figure 4). Subsequently, based on the expression of 1554 DEGs, correlation analysis was performed on three replicates at each age. The results showed that the correlation of three replicates for each age was above 0.973 (Figure 5), indicating good repeatability. In order to further understand the differences in expression patterns among the five groups, cluster analysis was performed based on the average expression of DEGs at each age. The results of cluster analysis using DEGs showed that the process of royal jelly secretion by the mandibular gland (3–16 d) can be divided into three stages with different expression patterns: 3–6 d, 9–12 d, and 16 d (Figure 6).

#### DEGs Function and Pathway Annotation

To better understand the biological processes occurring in the mandibular gland during the secretion of royal jelly (3–16 d), DEGs were categorized into three GO categories: biological process (BP), cellular component (CC), and molecular function (MF). The DEGs in AmC-6 vs. AmC-3 were involved in 14 BP terms, such as organic acid transport, carboxylic acid transport, and ion transport; 5 CC terms, such as actin cytoskeleton, myosin complex, and cytoskeletal part; and 28 MF terms, such as lyase activity, transporter activity, and transmembrane transporter activity (Figure 7a). In AmC-9 vs. AmC-3, 42 GO terms were annotated, including 15 BP terms, such as chitin metabolic process, amino sugar metabolic process, and glucosamine-containing compound metabolic process; 1 term (extracellular region) related to CC; and 26 MF terms, such as chitin binding, oxidoreductase activity, and structural constituent of the cuticle (Figure 7b). The DEGs in AmC-12 vs. AmC-3 were related to 10 BP terms, such as chitin metabolic process, amino sugar metabolic process, and glucosamine-containing compound metabolic process; 4 CC terms, such as extracellular region, cytoskeletal part, and actin cytoskeleton; and 35 MF terms, such as chitin binding, transporter activity, and ligand-gated ion channel activity (Figure 7c). A total of 56 GO terms were annotated in AmC-16 vs. AmC-3, covering 14 BP terms, such as chitin metabolic process, amino sugar metabolic process, and glucosamine-containing compound metabolic process; 9 CC terms, such as extracellular region, integral component of membrane, and plasma membrane; and 33 MF terms, such as chitin binding, receptor activity, and molecular transducer activity (Figure 7d).

Furthermore, KEGG pathway analysis showed that 11 pathways, including tyrosine, arachidonic acid, and ether lipid metabolism, were relevant to the DEGs in AmC-6 vs. AmC-3 (Figure 8a). In AmC-9 vs. AmC-3, 19 pathways were enriched, including tyrosine metabolism, propanoate metabolism, and extracellular matrix (ECM)–receptor interactions (Figure 8b). The DEGs in AmC-12 vs. AmC-3 were involved in eight pathways, including ECM–receptor interactions, pyrimidine metabolism, and glycerolipid metabolism (Figure 8c). In AmC-16 vs. AmC-3, the DEGs were related to 21 pathways, including ECM–receptor interactions, purine metabolism, and pentose and glucuronate interconversions (Figure 8d).

### 3.4. Verification of DEGS via RT-qPCR

*Actin* was selected as the internal reference gene, and 15 randomly selected genes with significant differences in expression between different ages were used for RT-qPCR validation (Figure 9). The results of a correlation analysis between the RNA-Seq and RT-qPCR data suggested that the expression trend of the RT-qPCR data was consistent with that of the transcriptome data (R^2^ = 0.953).

## 4. Discussion

In this study, we constructed an experimental population of *A. m. carnica* worker bees through single-male insemination. This facilitated the collection of worker bees with consistent genetic backgrounds. As the mandibular glands of bees are extremely small, a mixed sampling method was used for transcriptome sequencing. A limitation of this sampling method is that it is easy to lose the unique information of individuals. Generally, the natural mating mode of bees is poly-male fertilization, and the worker bees produced in a natural colony are half-sisters [32]. Worker bees with different genetic backgrounds may differ in their traits, behavior, and gene expression [33]. Using a single male to inseminate a queen bee to construct experimental populations can ensure consistency in the genetic background of the experimental samples and minimize errors caused by mixed samples.

In this study, electron microscopy was used to obtain images of the mandibular glands of worker bees at different ages. These photographs showed the tissue structure and cell ultrastructure of the mandibular glands 3–16 d after the worker bees emerged from their cells and provide a morphological basis for understanding the role of the mandibular glands in the secretion of royal jelly. According to these images, the mandibular glands had similar morphologies at different ages (3, 6, 9, 12, and 16 d), indicating that the development of the mandibular glands was completed by day 3. There were no drastic changes in tissue morphology during the subsequent developmental process. This mode differs from the development pattern of the hypopharyngeal glands (i.e., shrink, plump, and shrink) during the same period [20,34]. In addition, a large number of mitochondria and endoplasmic reticula were present in mandibular gland cells at different ages. The primary function of mitochondria is to provide energy for various cellular activities and regulate cellular metabolism, whereas the endoplasmic reticulum is the primary organelle involved in protein synthesis and transport. The metabolic level of a cell can be determined by the number of mitochondria and endoplasmic reticula. In this study, each age group contained abundant mitochondria and endoplasmic reticula, indicating that the mandibular gland already had a high metabolic level at day 3, and the metabolic level remains consistently high during the secretion of royal jelly. In addition, a small number of autophagic lysosomes were observed. Autolysosomes are formed through the fusion of the outer membrane of autophagosomes with lysosomes. Autophagy is an intracellular degradation system for breaking down certain cytoplasmic components, such as abnormal proteins, excess organelles, and invading microorganisms. Recent studies have also found that, in addition to clearing cellular contents, autophagy is related to fat synthesis in the body [35,36]. After knocking out the autophagy-related gene ATG7 in mice, the amount of white adipose tissue was reduced, adipocytes became smaller, and the number of lipid droplets in the cytoplasm increased [37]. Lipids in royal jelly are mainly synthesized by the mandibular glands. Autophagic lysosomes may be involved in the regulation of lipid metabolism in mandibular glands. However, the underlying mechanisms require further investigation.

Although the condition of the mandibular glands from 3 to 16 d could not be classified based on morphological changes, cluster analysis of the DEGs in the 3, 6, 9, 12, and 16 d old bees showed that the development of the mandibular glands from 3 to 16 d could be roughly divided into three categories. These three periods of differential gene expression may correspond to the three periods in which worker bees have different social divisions of labor as they grow older [38]. The 3 and 6 d genes were grouped together. At this stage, worker bees are primarily responsible for cleaning cells and feeding large larvae. The 9 and 12 d genes showed consistent expression patterns. At this stage, the worker bees secrete royal jelly and feed small larvae and the queen. Generally, royal jelly secretion peaks during this period (from 6 to 12 d) [39]. At 16 d, worker bees are mainly responsible for making honeycombs and processing nectar to produce honey.

The mandibular gland is the main organ secreting the fatty acids in royal jelly. A large proportion of lipids are synthesized in the mandibular glands and secreted into royal jelly. Lipid transport plays an important role in this process [26]. Therefore, we focused on the DEGs related to lipid transport at different ages. Compared with the number of genes upregulated at day 3, 1 (GB46277), 2 (GB49544 and GB46277), 1 (GB49869), and 0 lipid-transport-related genes were significantly upregulated at days 6, 9, 12, and 16, respectively, while 1 (GB48228), 1 (GB48228), 5 (GB48228, GB50662, GB52466, GB52465, and GB52464), and 4 (GB48228, GB50662, GB54507, and GB52465) lipid-transport-related genes’ expression levels had decreased significantly. The above results show that the lipid transport activity of the mandibular gland after 3 d did not increase with age and even showed a downward trend after 12 and 16 d.

In this study, we focused on the ECM–receptor interaction pathway, which showed significant enrichment in comparisons at different ages based on the gene expression and annotation results. The extracellular matrix (ECM) is a network of various macromolecules surrounding cells that participate in the regulation of cellular activities. The ECM supports cells mechanically. Cell surface receptors recognize several ECM macromolecules. These receptors are structurally homogeneous and classified into a single integrin superfamily. Integrins enable direct connections between cytoskeletal proteins and the ECM [40]. ECM–receptor interactions can induce signal transduction. The KEGG enrichment analysis conducted in this study showed that the ECM–receptor interaction pathway in the mandibular glands of worker bees mainly involves two upregulated genes, synaptic vesicle glycoprotein 2C (GB41126) and synaptic vesicle glycoprotein 2B (GB41127), and five downregulated genes, integrin beta-PS (GB42215), collagen alpha-l (IX) chain (GB52194), collagen alpha-1 (IV) chain (GB45968), collagen alpha-5 (IV) chain (GB45943), and synaptic vesicle glycoprotein 2B (GB41065). The ECM–receptor interaction pathway was significantly enriched in multiple comparison combinations (AmC-9 vs. AmC-3, AmC-12 vs. AmC-3, and AmC-16 vs. AmC-3), indicating that it plays an important role in the developmental process of the mandibular glands and in the secretion of royal jelly from 9 to 16 d. Studies have shown that the ECM plays an important role in adipogenesis and that the interaction between ECM components and adipocyte transmembrane receptors may affect the production of specific adipocytes [40,41]. Therefore, we speculate that the ECM–receptor interaction pathway plays a role in the synthesis and secretion of fatty acids from the mandibular glands of worker bees. However, this pathway is complex and involves a variety of cellular activities; its role in the mandibular glands of bees may be much greater.

## 5. Conclusions

Our research shows that the development of the mandibular glands of *A. m. carnica* is complete before 3 d, which is earlier than that of the hypopharyngeal glands. There were no obvious differences in the morphology of the mandibular glands or the number of organelles at different ages after the worker bees emerged from their cells. Overall, the mandibular gland was in a stable state during royal jelly secretion. The development of the mandibular glands from 3 to 16 d can be roughly divided into three stages according to the expression patterns of the DEGs. The ECM–receptor interaction pathway plays an important role in this process. These results elucidate the genetic basis of the involvement of the mandibular gland in the process of royal jelly secretion in *A. mellifera* and provide a reference for the genetic improvement of bees with high royal jelly production in the future.

## Figures and Tables

**Figure 1 insects-15-00176-f001:**
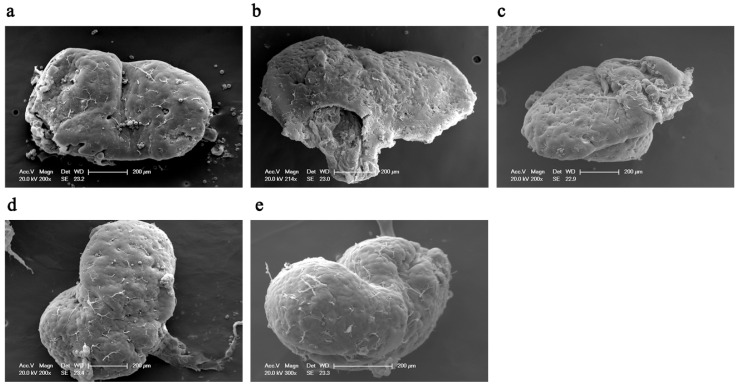
Scanning electron microscopy images of mandibular glands of 3, 6, 9, 12, and 16 d old worker bees. The scaleplate is 200 μm, and the magnification factor is 200×. (**a**–**e**) represent 3, 6, 9, 12, and 16 d, respectively.

**Figure 2 insects-15-00176-f002:**
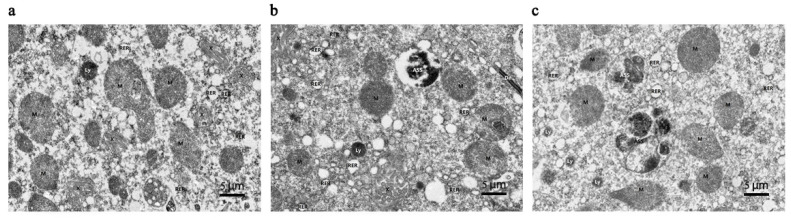
Transmission electron microscopy images of mandibular glands of 3, 6, 9, 12, and 16 d worker bees. The scaleplate is 5 μm, and the magnification factor is 6000×. (**a**–**e**) represent 3, 6, 9, 12, and 16 d, respectively. Notes: M (mitochondria), RER (rough endoplasmic reticulum), Ly (lysosome), ASS (autophagolysosome), Go (golgi), De (desmosome) and X (intracellular transport ducts).

**Figure 3 insects-15-00176-f003:**
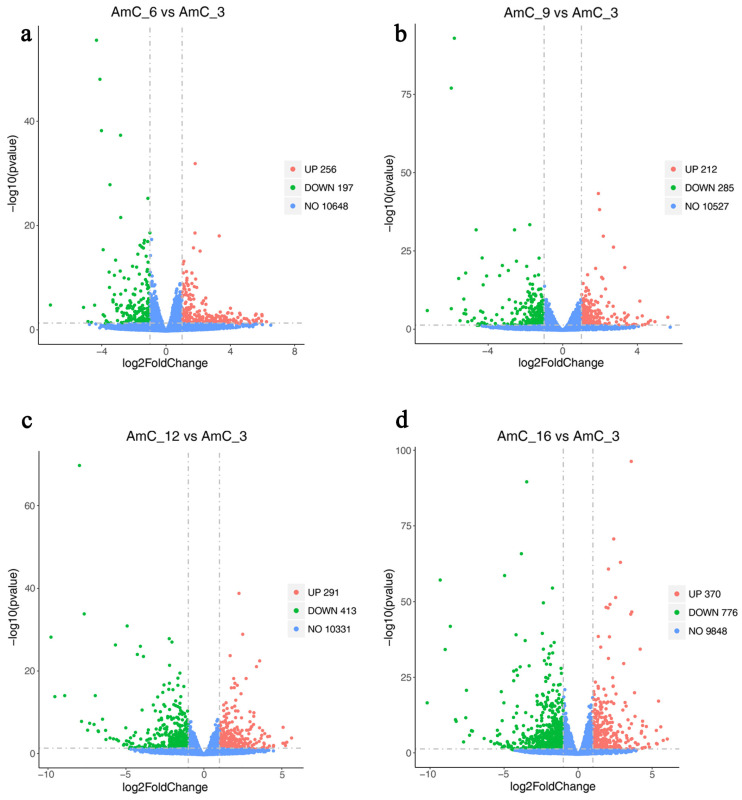
Volcano plot of differentially expressed genes (DEGs). (**a**) DEGs in AmC-6 vs. AmC-3. (**b**) DEGs in AmC-9 vs. AmC-3. (**c**) DEGs in AmC-12 vs. AmC-3. (**d**) DEGs in AmC-16 vs. AmC-3.

**Figure 4 insects-15-00176-f004:**
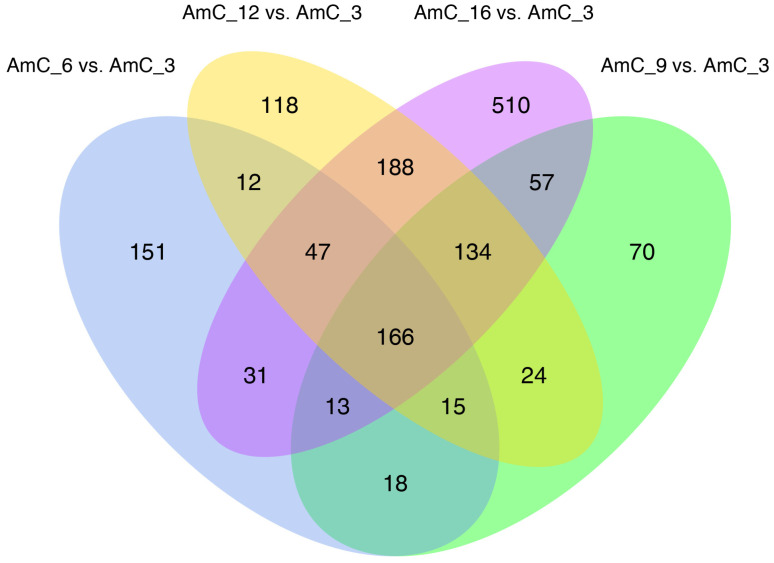
Venn analysis of DEGs in AmC-6 vs. AmC-3, AmC-9 vs. AmC-3, AmC-12 vs. AmC-3, and AmC-16 vs. AmC-3.

**Figure 5 insects-15-00176-f005:**
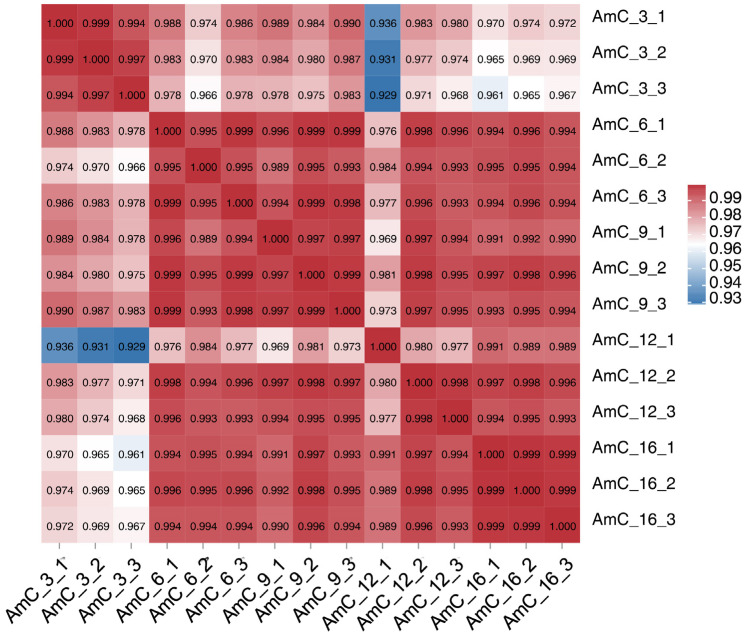
Heatmap of the correlations among three replicates at each age.

**Figure 6 insects-15-00176-f006:**
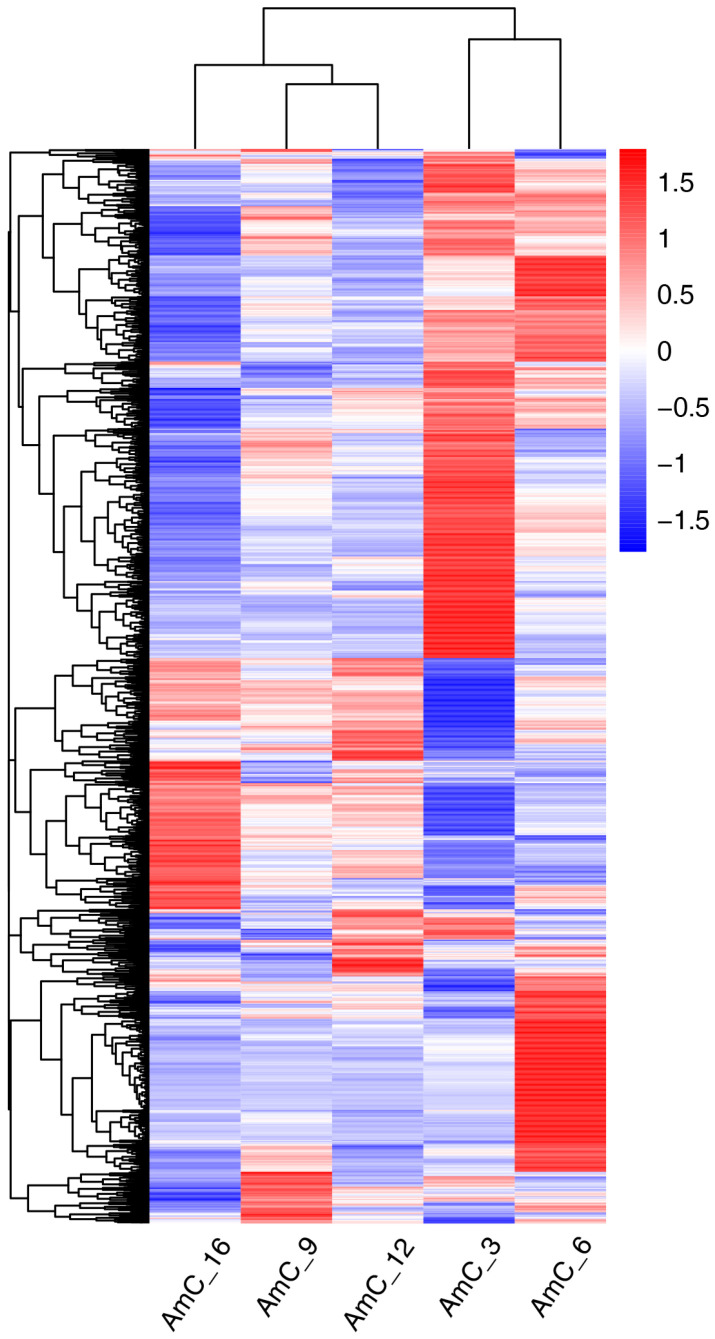
Heatmap of the average expression of DEGs in the mandibular glands of 3, 6, 9, 12, and 16 d old worker bees.

**Figure 7 insects-15-00176-f007:**
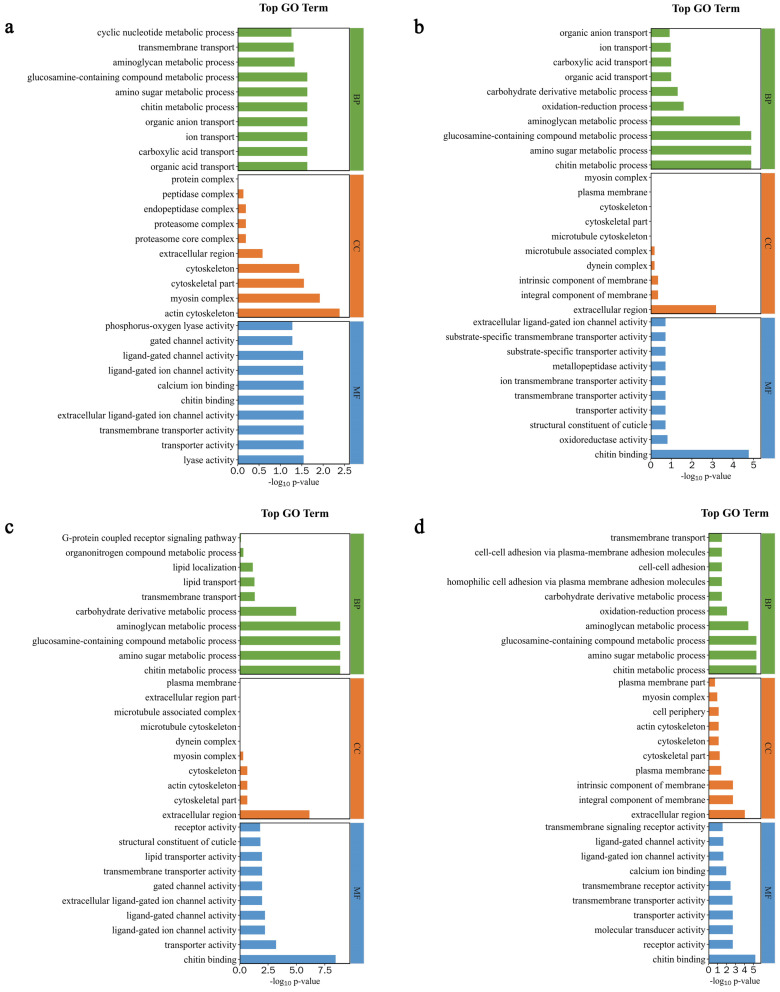
Bar chart of the top 10 terms in the Gene Ontology (GO) category. (**a**) GO terms in AmC-6 vs. AmC-3. (**b**) GO terms in AmC-9 vs. AmC-3. (**c**) GO terms in AmC-12 vs. AmC-3. (**d**) GO terms in AmC-16 vs. AmC-3.

**Figure 8 insects-15-00176-f008:**
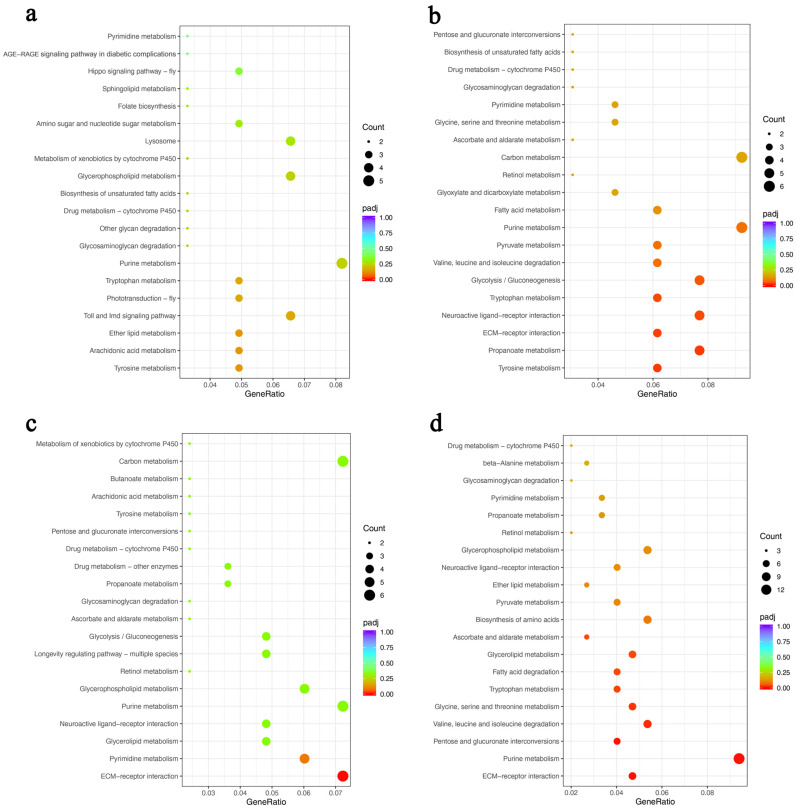
Bubble plot of the top 20 Kyoto Encyclopedia of Genes and Genomes (KEGG) pathways. (**a**) KEGG pathways in AmC-6 vs. AmC-3. (**b**) KEGG pathways in AmC-9 vs. AmC-3. (**c**) KEGG pathways in AmC-12 vs. AmC-3. (**d**) KEGG pathways in AmC-16 vs. AmC-3.

**Figure 9 insects-15-00176-f009:**
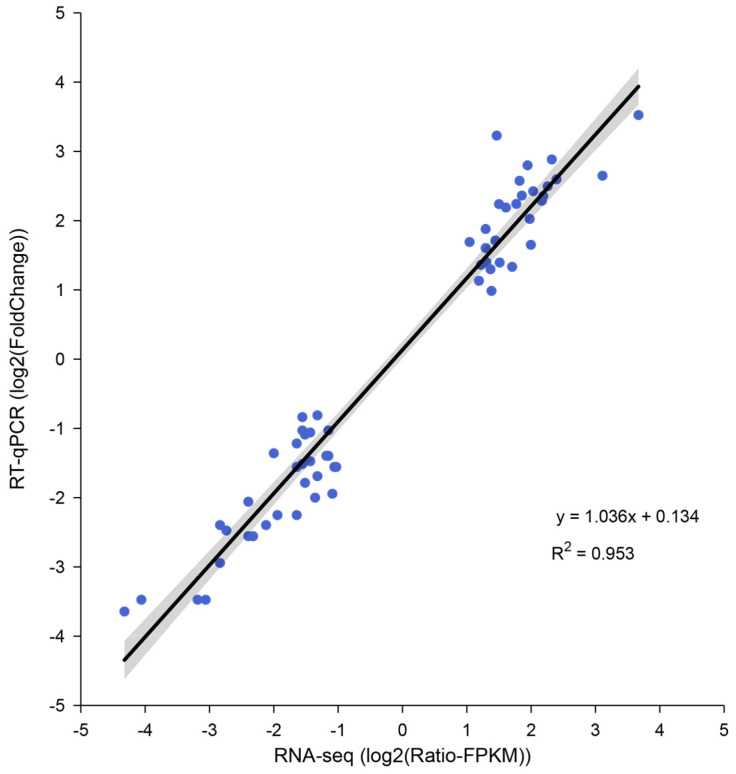
The correlation between the RNA-Seq and RT-qPCR data.

**Table 1 insects-15-00176-t001:** Detailed information on the raw reads of 15 samples.

Sample	Raw Reads	Clean Reads	Clean Bases	Q20	Q30
AmC_3_1	43,925,926	43,006,096	6.45 G	97.73	93.48
AmC_3_2	57,984,684	57,254,280	8.59 G	97.80	93.54
AmC_3_3	60,166,614	58,746,146	8.81 G	97.73	93.43
AmC_6_1	59,208,756	57,797,970	8.67 G	97.80	93.69
AmC_6_2	57,649,114	56,128,514	8.42 G	97.82	93.71
AmC_6_3	63,687,014	61,954,796	9.29 G	97.80	93.67
AmC_9_1	58,043,056	56,465,200	8.47 G	97.76	93.54
AmC_9_2	52,090,762	50,420,902	7.56 G	97.69	93.46
AmC_9_3	54,290,204	49,824,086	7.47 G	97.73	93.59
AmC_12_1	57,338,498	56,223,708	8.43 G	97.77	93.30
AmC_12_2	50,767,738	49,313,272	7.40G	97.54	93.12
AmC_12_3	66,511,356	62,514,040	9.38 G	97.59	93.27
AmC_16_1	56,240,840	54,560,816	8.18 G	97.66	93.36
AmC_16_2	60,768,102	58,984,280	8.85 G	97.59	93.19
AmC_16_3	51,563,118	50,354,212	7.55 G	96.89	91.61

## Data Availability

The sequencing data of the 15 samples in this study have been submitted to the NCBI Short Read Archive (SRA) under the BioProject accession number PRJNA1065419.

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
