# Peer review of "Revealing the Development Patterns of the Mandibular Glands of Apis mellifera carnica Based on Transcriptomics and Morphology"

_insects, 2024, doi:10.3390/insects15030176_

Round 1

Reviewer 1 Report

Comments and Suggestions for Authors

The paper provides an important characterization of mandibular gland development and metabolism. 

The experimental work undertaken appears to be technically sound. The presentation of transcriptomic data, mechanisms involved in royal jelly secretion and different expression patterns appeared to be accurate.

I think that authors provide quite interesting data. There are some minor questions which should be addressed before publication.

Introduction

Line 51-59. Please, add references. Is the royal jelly increase data relative to monthly production? Please, clarify.

Materials and Methods

Section 2. Line 92-98. Please, add how many colonies were used to collect the bees.

Discussion

Line 289-290 . “At 16 d, worker bees are mainly responsible for mak-289 ing honeycombs, collecting pollen and nectar, and defending the hive.” Please, change to “At 16 d, worker bees are mainly responsible for making honeycombs and process nectar to produce honey”.

Generally, bees with more than 20 days are responsible by collecting pollen and nectar, and defending the hive.

Author Response

Dear Reviewer,

We would like to thank the reviewer for carefully reading our manuscript (ID: insects-2862540). We appreciated the reviewer's constructive and insightful comments. On the next pages, our point-to-point responses to the queries raised by the reviewers are listed. All the changes have been highlighted in yellow in the revised manuscript.

Comment 1: Line 51-59. Please, add references. Is the royal jelly increase data relative to monthly production? Please, clarify.

Response: The royal jelly increase data here is not relative to monthly production. It is the data of an actual production process (72h). To avoid ambiguity, we have changed the description of royal jelly production performance to the more common annual production and added references in line 57-59.

Comment 2: Line 92-98. Please, add how many colonies were used to collect the bees.

Response: Thanks for your advice. A total of six colonies were used in this study. We have added it in line 92-93.

Comment 3: Line 289-290. “At 16 d, worker bees are mainly responsible for making honeycombs, collecting pollen and nectar, and defending the hive.” Please, change to “At 16 d, worker bees are mainly responsible for making honeycombs and process nectar to produce honey”.

Response: Thanks for your advice. We have revised the description in the manuscript in line 298-299.

We tried our best to improve the manuscript and we appreciate for Reviewer’s warm work earnestly and hope that the correction will meet with approval. Once again thank you very much for your comments and suggestions.

Reviewer 2 Report

Comments and Suggestions for Authors

Thank you for the opportunity to review the manuscript titled, Revealing the Development Patterns of the Mandibular Glands of Apis mellifera carnica Based on Transcriptomics and Morphology, by C. Pan et al.

 Summary: This manuscript examines age-related structural and transcriptional changes in the mandibular gland of Apis mellifera carnica. This study provides new information about the mandibular gland of honey bees that translates to the biology of nutrient production in royal jelly. These results could provide insight into honey bee management and in commercial interests in royal jelly production. The manuscript is well organized and logically presented. Confirmation of a subset of differentially-expressed genes using an additional method (RT-qPCR) is a strength of this study. Given the worldwide importance of native pollinators and declining insect numbers, as well as commercial interests in royal jelly, this report is timely and relevant.

 Major concern:

1.    The results presented do not identify significant time-dependent differences in either the scanning or transmission electron microscopy of the glands examined. Without significant differences at the timepoints shown, it is not clear that the scanning and transmission data is needed at each of the individual times listed.

 Minor concerns:

1.    Line 101 indicates that for transcriptome studies there were “three repetitions of 30 bees”. This information suggests there were three samples consisting of a pool of 30 bees for each time point. If this is the case, the authors should indicate that information explicitly. If this is not correct, the method should be clarified.

2.    Line 166: the nucleus is listed as a structural component identified in the electron micrograph but the nucleus does not appear in any of the images.

3.    In Figure 2d and 2e, there appear to be halos or clear areas around the mitochondria in several examples but not at all time points. Since this observation does not appear to be present in samples at each time examined, the authors should address this and confirm this is not due to artifact.

4.    Figure 3: The font in the legends should be enlarged.

5.    Figure 4: The legend should contain more detail regarding the groups being compared.

Figure 6: The labels on the X-axis have a small font that is difficult to read.

Author Response

Dear Reviewer,

We would like to thank the reviewer for carefully reading our manuscript (ID: insects-2862540). We appreciated the reviewer's constructive and insightful comments. On the next pages, our point-to-point responses to the queries raised by the reviewers are listed. All the changes have been highlighted in yellow in the revised manuscript.

Comment 1: Line 101 indicates that for transcriptome studies there were “three repetitions of 30 bees”. This information suggests there were three samples consisting of a pool of 30 bees for each time point. If this is the case, the authors should indicate that information explicitly. If this is not correct, the method should be clarified.

Response: Thanks for your kind advice. Each sample used for transcriptome studies is a pool of 30 bees. We have added the description of the samples at each time point in line 101.

Comment 2: Line 166: the nucleus is listed as a structural component identified in the electron micrograph but the nucleus does not appear in any of the images.

Response: Thanks for pointing out the mistake. We have removed the list of the nucleus in the notes of Figure 2.

Comment 3: In Figure 2d and 2e, there appear to be halos or clear areas around the mitochondria in several examples but not at all time points. Since this observation does not appear to be present in samples at each time examined, the authors should address this and confirm this is not due to artifact.

Response: We speculate that this phenomenon is caused by the difference of electron density between the mitochondria and the surrounding areas. While observing this phenomenon, we can also observe that the colors of the mitochondria in Figure 2d and 2e are darker than in other pictures. Overall, this phenomenon has little impact on our observations of organelles.

Comment 4: Figure 3: The font in the legends should be enlarged.

Response: Thanks for your advice. We have enlarged the font in the legends of Figure 3.

Comment 5: Figure 4: The legend should contain more detail regarding the groups being compared. Figure 6: The labels on the X-axis have a small font that is difficult to read.

Response: Thanks for your kind advice. We have supplemented the legend of Figure 4 and changed the form of Figure 7 to enlarged the information for easier reading.

We tried our best to improve the manuscript and we appreciate for Reviewer’s warm work earnestly and hope that the correction will meet with approval. Once again thank you very much for your comments and suggestions.